# The Role of Fractalkine in the Regulation of Endometrial Iron Metabolism in Iron Deficiency

**DOI:** 10.3390/ijms24129917

**Published:** 2023-06-08

**Authors:** Edina Pandur, Ramóna Pap, Gergely Jánosa, Adrienn Horváth, Katalin Sipos

**Affiliations:** 1Department of Pharmaceutical Biology, Faculty of Pharmacy, University of Pécs, H-7624 Pécs, Hungary; pap.ramona@pte.hu (R.P.); janosa.gergely@gytk.pte.hu (G.J.); adrienn.horvath2@gytk.pte.hu (A.H.); katalin.sipos@aok.pte.hu (K.S.); 2National Laboratory on Human Reproduction, University of Pécs, H-7624 Pécs, Hungary

**Keywords:** fractalkine, chemokine, iron deficiency, endometrium, haptoglobin, heme, iron metabolism

## Abstract

Iron is a crucial element in the human body. Endometrial iron metabolism is implicated in endometrium receptivity and embryo implantation. Disturbances of the maternal as well as the endometrial iron homeostasis, such as iron deficiency, can contribute to the reduced development of the fetus and could cause an increased risk of adverse pregnancy outcomes. Fractalkine is a unique chemokine that plays a role in the communication between the mother and the fetus. It has been demonstrated that FKN is involved in the development of endometrial receptivity and embryo implantation, and it functions as a regulator of iron metabolism. In the present study, we examined the effect of FKN on the iron metabolism of HEC-1A endometrial cells in a state of iron deficiency mediated by desferrioxamine treatment. Based on the findings, FKN enhances the expression of iron metabolism-related genes in iron deficiency and modifies the iron uptake via transferrin receptor 1 and divalent metal transporter-1, and iron release via ferroportin. FKN can activate the release of iron from heme-containing proteins by elevating the level of heme oxygenase-1, contributing to the redistribution of intracellular iron content. It was revealed that the endometrium cells express both mitoferrin-1 and 2 and that their levels are not dependent on the iron availability of the cells. FKN may also contribute to maintaining mitochondrial iron homeostasis. FKN can improve the deteriorating effect of iron deficiency in HEC-1A endometrium cells, which may contribute to the development of receptivity and/or provide iron delivery towards the embryo.

## 1. Introduction

Iron is a crucial element in the human body. It is implicated in oxygen delivery, ATP synthesis, DNA synthesis and repair [1]. Iron also functions as a cofactor in cytoprotective enzymes such as peroxidases and catalases [2]. Iron metabolism is controlled by the hepcidin–ferroportin axis, in which hepcidin is the regulator of iron release from the cell via the ferroportin iron exporter. Upon hepcidin binding, ferroportin is blocked, which inhibits further iron efflux, causing iron retention in the cells and decreasing serum iron concentration [3]. Hepcidin synthesis is regulated by many factors. At the transcriptional level, preprohepcidin mRNA synthesis is regulated by iron via the human hemochromatosis protein/transferrin receptor 1/transferrin receptor 2 (HFE/TfR1/TfR2) iron sensor system and the bone morphogenetic protein/bone morphogenetic protein receptor/SMAD (BMP/BMPR/SMAD) signaling pathway [4]. Inflammation is the strongest activator of preprohepcidin transcription via the Janus kinase/STAT (JAK/STAT) signaling pathway [5]. On the other hand, erythropoiesis suppresses hepcidin synthesis through erythroferrone [6].

During pregnancy, the iron supply of the developing embryo hinges completely on the mother [7]. Therefore, hepcidin synthesis is suppressed during pregnancy, especially in the second and third trimesters, probably by a regulator released by the placenta [3]. After fertilization, but before the development of the placenta, the endometrium has a pivotal role in the transport of nutrients towards the embryo. Iron delivery may be mediated by the ferroportin iron exporter expressed by endometrium cells. Alternatively, the release of heme or hemoglobin from the endometrial cells may contribute to the iron supply of the embryo [8]. Endometrium-derived extracellular vesicles are also supposed to play a part in iron sources [9].

It has been revealed that the mother’s iron status affects fertility as well, by contributing to the development of endometrial receptivity. Therefore, iron deficiency may decrease the chance of conception and cause infertility [10,11,12].

Maternal iron deficiency affects the iron availability of the embryo, causing an increased risk of adverse pregnancy outcomes such as mental disability, prematurity and lethality [13,14]. It has been proven that anemia during pregnancy reduces fetal development [7,15].

Fractalkine (CX3CL1; FKN) is a unique chemokine that plays a role in the communication between the mother and the fetus [16,17]. Fractalkine exists in two forms: a membrane-bound form with a function in cell–cell interactions, and a soluble form with a role in the regulation of intracellular signaling processes. Both types of FKN can interact with the FKN receptor, CX3CR1 [18,19]. FKN is expressed by the endometrium cells, while both endometrium cells and embryonic trophoblasts express CX3CR1. The FKN/CX3CR1 axis regulates the mitogen-activated protein kinase (MAPK) pathway implicated in cell proliferation, growth and differentiation, the phosphoinositide 3-kinase-Akt serine/threonine kinase/Nuclear factor kappa-light-chain-enhancer of activated B cells (PI3K/Akt/NFkB) pathway, affecting apoptosis, protein synthesis and the transcription of several pro-inflammatory cytokines (e.g., IL-1, IL-6, IL-8 and TNFα), and the phospholipase C/Calcium (PLC/Ca) signal, regulating chemotaxis [20,21,22].

It has been demonstrated that FKN is involved in the development of endometrial receptivity and embryo implantation, and it has a role in pregnancy [23], although its role in pregnancy is not fully understood [24,25,26]. In addition, it has been described that FKN may act as a regulator of iron metabolism in macrophages, microglia and endometrium cells [8,27]. Recent findings revealed that FKN improves the expression of receptivity-related genes in iron deficiency [28].

In the present study, we focused on the effect of FKN on the iron metabolism of iron deficient HEC-1A endometrial cells mediated by desferrioxamine (DFO) treatment. We examined the iron uptake, and release as well as iron storage and the mitochondrial iron utilization in iron deficiency in the presence of FKN. The modulating effect of the iron supplementation on FKN-treated cells was also monitored. Based on the findings, it seems that FKN can improve the deteriorating effect of iron deficiency in HEC-1A endometrium cells. FKN causes iron redistribution and modifies iron utilization of the endometrium cells, which may contribute to receptivity and/or provide iron delivery towards the embryo at the early stage of pregnancy.

## 2. Results

### 2.1. Iron Deficiency Affects the Expression of the Iron-Related Genes in HEC-1A Endometrium Cells

We examined the effect of iron deficiency on iron metabolism-related genes in endometrium cells at the mRNA level. The mRNA expression of the iron importers transferrin receptor 1 (TfR1) and divalent metal transporter (DMT-1) altered in different ways. The TfR1 mRNA did not show a reduction but elevated after a 72 h-long DFO treatment, suggesting the decreasing iron pool and the developing iron deficiency in the cells (Figure 1A).

The mRNA level of DMT-1 significantly decreased in the endometrium cells, suggesting a reduced need for iron transport from the endosomes to the cytosol (Figure 1A). In the meantime, the mRNA expression level of the iron exporter ferroportin (FP) showed a significant downregulation compared to the control cells, reflecting the depleted iron content of the cells (Figure 1A).

The mRNA level of the cytosolic iron storage protein ferritin heavy chain (FTH) was also significantly lowered, proving the decreasing rate of iron storage or the increasing rate of iron liberation from ferritin (Figure 1B). The expression of the hepcidin coding gene, HAMP, significantly decreased at each time point (Figure 1C).

The expression of the mitochondrion-related iron genes revealed less dramatic changes. The ferrochelatase (FECH) mRNA level reduced at 24 h and 48 h, but at 72 h of DFO treatment, its level elevated close to the control level (Figure 1D). In the case of heme oxygenase-1 (HO-1), significant downregulation was observed only at 24 h DFO treatment (Figure 1D). Moreover, the gene expression level of the mitochondrial iron transporters mitoferrin-1 and mitoferrin-2 (Mfrn-1 and 2) was not altered significantly in iron deficiency, suggesting that the mRNA expression of these transporters does not hinge on the iron availability of the cells (Figure 1F).

The gene expression of haptoglobin, which is implicated in the extracellular transport of heme/hemoglobin, was significantly downregulated at each time point (Figure 1E).

### 2.2. Fractalkine Alters the mRNA Expression Levels of the Iron Transporters Transferrin Receptor 1, Divalent Metal Transporter-1 and Ferroportin

To reveal the effect of FKN on the expression of iron transporters in iron deficient HEC-1A cells, we examined the gene expression of TfR1, DMT-1 iron importers and FP, the only known iron exporter. The 10 ng/mL FKN significantly triggered the expression of TfR1 after 24 h DFO treatment, both in a serum-free and a serum-supplemented culture media, compared to both DFO and DFO + F20 treatments (Figure 2A). At 48 h of DFO treatment, FKN provided an opposite effect; it decreased the TfR1 mRNA level compared to the DFO treatment in a serum-free environment, suggesting an inhibitory effect on the transcription of TfR (Figure 2B). When transferrin-bound iron was reintroduced to the cells via serum supplementation, FKN did not modify the TfR1 mRNA level (Figure 2B).

In the case of DMT-1, the 10 ng/mL of FKN was able to successfully elevate the DMT-1 mRNA level compared to the control, DFO and DFO + F20 treatments (Figure 2A). In the case of serum supplementation together with FKN treatment, we did not observe significant alterations in the DMT-1 mRNA expression (Figure 2A). After the 48 h-long DFO treatment, FKN further reduced DMT-1 gene expression, which was in unison with the TfR1 expression level (Figure 2B). When the culture medium was supplemented with serum for FKN treatment, only the 10 ng/mL of FKN decreased the DMT-1 mRNA level (Figure 2B).

FKN was capable of significantly increasing the FP mRNA level in a concentration-dependent manner in a serum-free medium after 24 h of DFO treatment (Figure 2C). Using a serum-containing medium, in the case of FKN treatment of the iron-deficient HEC-1A cells, the 10 ng/mL of FKN was more efficient in elevating FP mRNA level (Figure 2C). After the 48 h DFO treatment, FKN reduced the FP mRNA level in a serum-free culture medium (Figure 2D). Meanwhile, the presence of serum in FKN treatment modified its effect. However, the 10 ng/mL of FKN did not cause a significant change, while the 20 ng/mL of FKN significantly raised FP mRNA expression compared to DFO and F10 treatments (Figure 2D).

According to these results, FKN can modify the mRNA expression of the iron transporters, but this effect may depend on the duration of the iron deficiency. Moreover, the presence of iron in the cell cultures has an impact on the effect of FKN.

### 2.3. Fractalkine Alters the Protein Levels of the Iron Transporters Transferrin Receptor 1, Divalent Metal Transporter-1 and Ferroportin in Iron Deficiency

At the protein level, the 10 ng/mL of FKN significantly decreased the TfR1 level compared to the 24 h-long DFO treatment in a serum-free environment (Figure 3A,B). Meanwhile, the 20 ng/mL of FKN treatment significantly elevated the TfR1 protein level compared to both DFO and F10 treatments (Figure 3A,B). Moreover, the TfR1 protein level reached the control level using the higher concentration of FKN. When serum was utilized together with FKN, no significant change was observed (Figure 3A,B).

In the case of DMT-1, the 20 ng/mL of FKN treatment significantly reduced the protein level compared to DFO and DFO + F10 treatments in a serum-free culture medium, suggesting that FKN at higher concentration inhibits iron uptake through the plasma membrane or iron release from the endosomes (Figure 3A,C). In the presence of serum, FKN had the opposite impact on the DMT-1 protein level. Both FKN concentrations significantly raised the DMT-1 protein level, although 10 ng/mL of FKN was more effective on DMT-1 (Figure 3A,C).

The iron exporter FP level increased only in iron-deficient HEC-1A cells when 20 ng/mL of FKN was utilized in the experiments in a serum-free environment (Figure 3A,D). When the cells were supplemented with serum after 24 h of DFO treatment, the presence of iron was not able to increase the FP level, but both FKN concentrations significantly elevated the FP protein level compared to the DFO treatment (Figure 3A,D).

According to our observations, the presence of iron can modify the action of DFO treatment, but FKN is still capable of increasing the levels of the iron transporter proteins, suggesting its role in the regulation of iron metabolism.

The longer DFO treatment caused less dramatic changes in the protein expression levels of the iron transporters in a serum-free environment (Figure 4A–D). The 48 h-long DFO treatment reduced the TfR1 and DMT-1 levels but elevated the FP protein level compared to the controls (Figure 4A–D). The addition of FKN after DFO treatment did not affect the protein levels (Figure 4A–D).

Interestingly, serum supplementation together with 10 ng/mL of FKN resulted in the opposite alteration of all three iron transporters compared to the results of the 24 h DFO treatments: their protein levels significantly decreased in the presence of FKN compared to the DFO-only treatment (Figure 4A–D).

FKN at a higher concentration significantly increased the TfR1 and the DMT-1 protein levels compared to the controls, and DFO-only treatments (Figure 4A–C). Meanwhile, it increased the FP protein level compared to the control, but it was not able to elevate the FP level compared to the DFO treatment (Figure 4A,D).

These results suppose that the effect of FKN may depend on the iron status of the cells.

### 2.4. Fractalkine Acts Differently on the Hepcidin Gene (HAMP) and Ferritin Heavy Chain mRNA Expression Levels with or without Serum Supplementation

Hepcidin functions as an inhibitor of iron release from the cells; therefore, it may modify iron storage as well. We examined the hepcidin gene HAMP and FTH mRNA expression levels to reveal whether FKN can regulate their expression. Iron deficiency decreased both HAMP and FTH mRNA levels in 24 h and 48 h DFO treatments as well (Figure 5A–D).

The addition of FKN to the iron-deficient cells modified the mRNA expression levels. After 24 h DFO treatment, the 20 ng/mL of FKN further reduced the HAMP level, but after 48 h DFO treatment, FKN increased it. On the other hand, in the case of serum supplementation, the 10 ng/mL of FKN significantly increased the HAMP mRNA expression (Figure 5A,B).

The FTH mRNA level significantly decreased in the presence of FKN at both concentrations, but only after 24 h DFO treatment. In the case of the serum supplementation, similarly to the changes in HAMP mRNA level, the 10 ng/mL of FKN significantly increased the FTH mRNA expression (Figure 5C,D).

### 2.5. Fractalkine Affects the Level of the Iron Storage Protein Ferritin Heavy Chain

Iron deficiency significantly decreased the FTH level at the protein level, even if transferrin-bound iron (serum) was freshly available for the cells (Figure 6A–D). Without serum, FKN was not able to increase the FTH protein level (Figure 6A–D). After the addition of FKN together with serum after 24 h DFO treatment, the FTH protein level decreased compared to the control and DFO treatment (Figure 6A,B). After 48 h of DFO treatment, administration of FKN in the presence of serum significantly increased the FTH level compared to the DFO-only treatment (Figure 6C,D). Based on the results it seems that FKN can trigger the elevation of the FTH protein level only in the presence of iron.

### 2.6. Fractalkine Alters the mRNA Expression of the Genes Related to Mitochondrial Iron Utilization

Ferrochelatase (FECH) and heme oxygenase-a (HO-1) are implicated in heme synthesis and degradation, while mitoferrins (Mfrn-1 and -2) function as mitochondrial iron importers. We examined the mRNA expression levels of these genes involved in mitochondrial iron metabolism. In iron deficiency, FKN was not capable of restoring FECH mRNA expression both at 24 h and 48 h of DFO treatments (Figure 7A,B). When serum was added to the cultures together with FKN, only 10 ng/mL of FKN was successful in elevating the FECH protein level. In the case of HO-1, FKN significantly raised the HO-1 mRNA expression levels, following both DFO treatments and with or without the presence of serum, compared to the DFO-only treatments (Figure 7A,B).

The Mfrn-1 mRNA level decreased in the case of FKN treatment compared to DFO treatments (Figure 7C,D). In the meantime, the Mfrn-2 mRNA level increased in FKN treatment in a serum-free environment (Figure 7C,D). In the presence of serum, FKN was able to elevate Mfrn-1 mRNA expression after 24 h and 48 h DFO treatments in a concentration-dependent manner (Figure 7C,D). In the case of Mfrn-2, the FKN raised the Mfrn-2 mRNA level only after 24 h of DFO treatment (Figure 7C,D).

According to these observations, it seems that Mfrn-2 mRNA expression does not hinge on the iron availability of the cells.

### 2.7. Fractalkine Alters the Protein Levels of the Genes Related to Mitochondrial Iron Utilization after DFO Treatment

After the 24 h-long DFO treatment, FKN significantly decreased FECH protein level in both a serum-free and a serum-supplemented environment (Figure 8A,B). In the case of HO-1, we revealed the opposite results; FKN significantly elevated the HO-1 protein level compared to the DFO treatment (Figure 8A,C).

In the case of Mfrn-1 mitochondrial iron importer, FKN significantly reduced the Mfrn-1 protein level in a serum-free culture medium, but this effect was reversed in the presence of serum, where FKN significantly elevated the Mfrn-1 protein level compared to the 24 h DFO treatment (Figure 8A,D).

The protein level of the Mfrn-2 mitochondrial iron importer significantly increased at FKN treatment both with and without serum addition compared to the DFO-only treatment (Figure 8A,E).

After the 48 h DFO treatment, FKN significantly increased the FECH protein level in a serum-free environment but decreased it in the presence of serum (Figure 9A,B). We found the same changes in the case of the HO-1 protein level (Figure 9A,C). These observations may show that FKN acts as a regulator of FECH synthesis, but the iron availability of the cells can modify its action.

In the case of Mfrn-1, 20 ng/mL of FKN significantly reduced the Mfrn-1 protein level without serum addition (Figure 9A,D). In the presence of serum, it seemed that the effect of FKN depended on the concentration. The 10 ng/mL of FKN significantly elevated but the 20 ng/mL of FKN reduced the Mfrn-1 protein level compared to the DFO treatment (Figure 9A,D). In the case of the Mfrn-2 protein level, the effect of FKN was completely different.

In a serum-free culture medium, FKN significantly increased the Mfrn-2 level, and after the re-addition of iron, FKN treatment decreased it (Figure 9A,E).

Based on the results, it is supposed that the effect of FKN depends on the cellular iron content and/or the iron availability.

### 2.8. Fractalkine Changes the mRNA and Protein Levels of Haptoglobin

In our previous study, we observed that HEC-1A cells may transport the iron as heme towards the trophoblast cells. Therefore, we analyzed the expression of the heme transporter haptoglobin in iron deficiency. In a serum-free environment, FKN significantly elevated both the mRNA expression and the secretion of haptoglobin after 24 h DFO treatment (Figure 10A,C). After 48 h DFO treatment, FKN was not capable of increasing the haptoglobin level; moreover, the addition of FKN significantly decreased haptoglobin expression (Figure 10B,D). If FKN was added to the cells together with serum, the lower concentration of FKN increased, while the higher concentration of FKN reduced, the haptoglobin level after 24 h DFO treatment (Figure 10A,B). We observed the opposite results after 48 h DFO treatment: FKN significantly increased haptoglobin mRNA and protein levels in the presence of serum.

### 2.9. Alterations in the Total Iron Content and Heme Concentration of the Iron-Deficient HEC-1A Cells after Fractalkine Treatment

Our results showed that iron transport, storage and utilization were altered in iron deficiency and that FKN was able to modify the iron metabolism of the HEC-1A cells. We examined whether these observations are in accordance with the iron content of the endometrium cells.

We found that FKN significantly increased the total intracellular iron content of the HEC-1A cells in both serum-free and serum-containing culture media, though the iron content of the cells could not reach the control levels in the case of serum-supplemented FKN treatment (Figure 11A,B).

The heme concentration followed the changes in the total iron content of HEC-1A cells. FKN significantly increased the heme concentration in both serum-free and serum-containing culture media after 24 h and 48 h DFO treatments (Figure 11C,D).

Considering our observations, FKN causes iron redistribution in iron deficiency and modifies iron utilization of the HEC-1A endometrium cells.

## 3. Discussion

Maternal iron homeostasis affects both fertility and pregnancy [10,12]. Endometrial iron metabolism is also implicated in endometrium receptivity and embryo implantation. Moreover, at the early stages of pregnancy, the endometrium is responsible for providing nutrients for the growing embryo [7]. Disturbances of the maternal as well as the endometrial iron homeostasis can contribute to the reduced development of the fetus and could cause an increased risk of adverse pregnancy outcomes [14]. Iron deficiency affects the maturation and development of the fetal central nervous system by reducing synaptogenesis and synaptic plasticity and inhibiting the proliferation of neuronal precursors [15].

Fractalkine (FKN) is a chemokine, which acts on the fractalkine receptor (CX3CR1). Upon receptor binding, FKN activates three major signaling pathways: PI3K/Akt/NFκB, MAPK and PLC, regulating proliferation, growth, differentiation, apoptosis and chemotaxis [20,21,22]. The FKN/CX3CR1 interaction mediates maternal–fetal communication as well [16,17].

It has been revealed that FKN may contribute to embryo implantation, as well as to the development of endometrial receptivity, and plays a protective role in pregnancy [23]. Moreover, it regulates the iron transport between the endometrial cells and trophoblasts by modifying the expression of iron metabolism-related genes [8].

The endometrium is a heterogenous tissue containing multiple types of cells such as epithelial, glandular, stromal, vascular, and immune cells, all with different functions [29]. Moreover, the proportions of the different cell types vary with the female cycle. According to the histological examination, the endometrium is lined by a luminal epithelium containing tubular glands. These glands transmit through the stoma towards the myometrium [30]. The changes in the endometrium play a crucial role in the evolvement of receptivity and the implantation process [31]. The HEC-1A cells express endometrial cell surface markers and adhesion molecules, such as intercellular adhesion molecules (ICAM), mucin-1, moesin and ezrin, and secrete tissue inhibitors of matrix metalloproteinases [32,33,34]. HEC-1A cells also express both estrogen and progesterone receptors, as well as cytokines [28,32,33]. According to the gene expression profile, HEC-1A cells represent the phenotype of luminal or glandular epithelial cells [35].

In the present study, we investigated the effect of FKN on the iron metabolism of iron deficient HEC-1A endometrial cells. According to the gene expression analysis of iron-related genes, iron deficiency disturbed the iron homeostasis of the endometrium cells. The level of TfR1 did not decrease; moreover, it was elevated after long-term iron chelation, suggesting a compensatory mechanism, which is activated by the decreasing intracellular iron level [36]. DMT-1 showed a decreasing phenomenon, which may reflect the reduced iron uptake and release from the endosomes into the labile iron pool [37,38]. The ferritin heavy chain expression also decreased, proving the reduction of iron storage and the development of iron deficiency [39]. In parallel with this observation, the cells downregulated the level of FP, inhibiting the iron release from the cells. The decreasing level of HAMP, the hepcidin gene, may help in the restoration of iron uptake [40]. FECH mRNA expression, as well as HO-1 expression, decreased, suggesting a downregulation of heme metabolism, although the HO-1 level began to elevate at the late stage of the experiments. At this stage, HO-1 may trigger the recircularization of iron to maintain the basal iron-dependent functions of the cells (e.g., DNA synthesis, energy production). The drop in the haptoglobin expression underlies the reduced heme synthesis and release from the cells.

In contrast to the previous results, the mitochondrial iron transporters seemed to be less sensitive to the intracellular iron content, or to not be regulated by iron at the transcriptional level. These results obtained in endometrial cells with iron deficiency are in accordance with previous observations on iron-deficient hepatocytes, macrophages and cardiomyocytes [39,41,42].

The effect of FKN was examined both with and without serum on the iron-deficient HEC-1A cells. In the presence of serum, we were able to test whether the re-administration of iron can change the effect of FKN treatment.

In the case of the iron uptake, FKN was able to modify both TfR1 and DMT-1 levels at mRNA and protein levels; however, the duration of DFO treatment, as well as the presence of iron together with FKN, could reverse and/or modify their expression. Since the TfR1 mRNA level increased upon FKN treatment either in a serum-free or supplemented environment, we suppose that the FKN/CX3CR1 axis regulates its mRNA expression. It may be controlled by the NFκB signaling pathway, a known transcriptional regulator of TfR1 [43]. A delay between mRNA and protein synthesis of TfR1 was also observed. A similar phenomenon was found in the case of DMT-1, although the presence of serum seemed to be an activator factor of FKN action. The elevation in the protein levels could be increased by IRP1, which acts on the 3′-UPR of mRNA and enhances translation. IRP1 activity is known to be increased in iron deficiency, which triggers iron uptake [44]. In the meantime, it can be elevated in an iron-independent way, via the inflammatory response mediated by the NFκB signaling pathway [45]. Despite the aforementioned observations, a strong inhibitory effect of FKN on iron uptake protein levels was revealed at 48 h DFO treatment in the presence of serum. The reason for this result might be that iron supplementation decreases the activity of IRP1 on the translation of TfR1 and DMT-1 and/or that iron triggers FKN expression, which harms the CX3CR1 signaling [44,46].

Ferroportin (FP) mRNA expression was enhanced by FKN treatment, suggesting that, in shorter iron deficiency, FKN activates the liberation of iron from heme by HO-1, whose level was elevated by FKN, for maintaining the continuous iron export. Despite this result, the more severe iron deficiency overrides the action of FKN. In the presence of serum, the IRP-mediated upregulation of FP may also contribute to the elevated FP level [47]. According to the results, it is proposed that FKN/CX3CR1 functions as a transcriptional regulator, while IRP1 regulates the expression of the examined proteins at the post-transcriptional level.

It has been observed that DFO treatment, which evokes iron deficiency and mimics hypoxia, decreases fractalkine expression in endothelial cells [48]. Both circumstances act as negative regulators of HAMP (hepcidin) transcription, which was observed in our experiments [49]. FKN seems to act as a negative regulator of HAMP at less severe iron deficiency, while it notably enhances HAMP transcription at lower iron concentrations. In the first case, FKN may help in maintaining iron release from the cells, while in the case of severe iron deficiency, FKN assist in iron retention. These effects can be mediated through the NFκB and MAPK pathways controlled by the FKN/CX3CR1 axis [21,22]. Iron supplementation reactivates HAMP expression to cease the iron imbalance of the endometrium cells.

DFO decreases intracellular iron stores by chelating iron from the cytoplasmic labile iron pool and ferritin [50]. In the case of FTH protein, FKN decreased its level without the presence of iron but elevated it when serum supplementation was carried out together with FKN treatment. Although FKN provided a positive impact on the FTH level compared to the DFO treatment, which suggests that FKN acts as a modulator of FTH synthesis, these levels would not be able to reach the FTH level of the control cells. It cannot be ruled out that FKN provokes IRP1 activity via the NFκB pathway, which in turn stabilizes and promotes FTH translation [45].

Iron chelators may function as Nrf2 activators, which is also regulated by the FKN/CX3CR1 axis [8,47]. In our previous study, it was revealed that FKN can modify the Nrf2 protein level, as well as its negative regulator, Keap-1, in iron deficiency, which may cooperate with the impact of iron availability on the level of FP, FTH and HO-1 [51,52].

Mitochondrial iron homeostasis is essential in maintaining energy production, heme synthesis and iron-sulfur cluster synthesis [53]. FKN produces a more robust effect on heme degradation by increasing the level of HO-1, which may contribute to iron redistribution and/or iron release from the iron deficient endometrium cells. It is interesting that HEC-1A cells express both types of mitochondrial iron importers. Moreover, it seems that the expression of these transporters, mitoferrin-1 and 2 (Mfrn-1 and 2), is not dependent on the iron availability and content of the cells. On the other hand, FKN inhibits Mfrn-1 but prompts Mfrn-2 at the protein level without serum, whereas FKN elevated both importers at the protein level in the presence of serum, except in the case of Mfrn-2 after the longer DFO treatment. It also suggests that FKN/CX3CR1 signaling, rather than mitoferrins, is sensitive to intracellular iron concentration. For this reason, mitoferrins can maintain the mitochondrial iron transport from the cytosol to support mitochondrial iron homeostasis [53].

It has been proposed that iron is delivered from the endometrium towards the embryo via heme transport accomplished by endometrium-secreted haptoglobin [8]. In short-term iron deficiency, FKN increased both haptoglobin transcription and secretion, supporting our hypothesis. In the meantime, FKN was able to elevate the heme concentration of the iron-deficient HEC-1A endometrium cells, even without or with available iron in the culture medium. We have to note that even in the presence of serum, the heme levels did not reach or override the control levels. Evaluating the iron content of the endometrium cells, FKN treatment increased the amount of intracellular iron, suggesting a redistribution of iron that may originate from iron-containing proteins.

Considering that HEC-1A cells resemble the luminal and glandular epithelial cells, our results model the changes in the epithelial layer of the endometrium, which is implicated in the development of receptivity and the implantation of the embryo.

Our findings support the hypothesis that FKN functions as a regulator of the iron metabolism of HEC-1A endometrium cells. FKN enhances the expression of iron metabolism-related genes in DFO-mediated iron deficiency and modifies the iron uptake via TfR1 and DMT-1, and iron release via FP. FKN can activate the release of iron from heme-containing proteins by elevating the level of HO-1, contributing to the redistribution of intracellular iron content. It was revealed that the endometrium cells express both Mfrn-1 and Mfrn-2 and that their levels are not dependent on the iron availability of the cells. FKN may also contribute to maintaining mitochondrial iron homeostasis. Our observations also support the idea that that FKN enhances heme synthesis and may promote heme release and transport by haptoglobin towards the embryo.

Serum supplementation alters the outcome of FKN treatment. Hence, we hypothesized that FKN expression and/or regulation may hinge on the iron availability of the cells. however, we cannot exclude the possibility that not only iron but also additional serum components (e.g., cytokines) are responsible for these changes. Although our observations allow a deeper understanding of the regulation of iron metabolism in iron deficient endometrial cells, the utilization of an in vitro cell culture model is the main limitation of the study. The findings described above should be verified in an in vivo animal or a human study.

## 4. Materials and Methods

### 4.1. Cell Cultures and Treatments

The human endometrial cell line HEC-1A (ATCC HT-112) was maintained in McCoy’s 5A medium (Ishikawa and Grace modification; Corning Inc., Corning, NY, USA) supplemented with 10% fetal bovine serum (FBS, EuroClone S.p.A, Pero, Italy) and 1% penicillin/streptomycin (P/S, Lonza Ltd., Basel, Switzerland). Desferal (desferrioxamine; DFO) was purchased from Novartis (Novartis Hungária Kft., Budapest, Hungary). The DFO was dissolved in distilled water, making 100 mM of stock solution. The cells were treated with 100 μM DFO for 24 h, 48 h or 72 h in a serum-free culture medium to develop iron deficiency. Iron deficiency was proven via the heme content of the cells (Appendix A). After triggering iron deficiency for 24 h and 48 h, the cell cultures were separated into two groups. The first group was supplemented with recombinant human fractalkine (Shenandoah Biotechnology Inc., Warwick, PA, USA) at 10 and 20 ng/mL concentrations in a serum-free environment for 24 h. The second group was treated with 10 and 20 ng/mL fractalkine together with serum supplementation for 24 h. In the second case, iron was newly available for the cells. In each experiment, untreated cells were used as controls (Figure 12). The control cells were incubated for the same time (24 h, 48 h and 72 h) in a serum-free environment as the DFO-treated cells and were used as controls at the corresponding time point. The cells were cultured in a humified atmosphere in the presence of 5% CO_2_ at 37 °C.

### 4.2. Real-Time PCR

The cells were seeded onto 6-well plates (Biologix Europe, Hallbergmoos, Germany) using 5 × 10^5^ cells/well. Then, the cultures were treated as described earlier. After incubation, the cells were collected via trypsinization and pelleted using centrifugation. The pellets were washed once with 1 × phosphate-buffered saline (PBS, Lonza Ltd., Basel, Switzerland). Total RNA was isolated using an Aurum Total RNA Mini Kit (Bio-Rad Inc., Hercules, CA, USA). The RNA concentration of each sample was determined using a MultiSkan GO spectrophotometer (Thermo Fisher Scientific Inc., Waltham, MA, USA). The cDNA synthesis was performed using the iScript Select cDNA Synthesis Kit (Bio-Rad Inc., Hercules, CA, USA). The real-time PCR reactions were carried out in a CFX96 Opus Real-Time PCR System (Bio-Rad Inc., Hercules, CA, USA) using a SYBR Green reagent (iTaq Universal SYBR Green Reagent Mix; Bio-Rad Inc., Hercules, CA, USA). The relative mRNA expression level (fold change) of the target gene was calculated with the Livak (∆∆Ct) method using the Bio-Rad CFX Maestro 2.3. software (Bio-Rad Inc., Hercules, CA, USA). Glyceraldehyde 3-phosphate dehydrogenase (GAPDH) housekeeping gene was used for the normalization of the real-time PCR reactions. The mRNA expression levels of the control cells were regarded as 1. The nucleotide sequences of the primers used in the experiments are presented in Table 1. The real-time PCR analyses were carried out in triplicate in three independent experiments.

### 4.3. Western Blotting

The HEC-1A cells were cultured in 6 cm-wide cell culture dishes (Biologix Europe, Hallbergmoos, Germany) using 10^6^ cells/dish. After the treatments, the cells were harvested via centrifugation, lysed in 180 μL of lysis buffer containing 50 mM Tris-HCl, 150 mM NaCl, 0.5% Triton X-100, pH 7.4 and supplemented with 20 μL of Complete Mini protease inhibitor cocktail (Roche Ltd., Basel, Switzerland). The protein content of the samples was determined from a photometric measurement using DC Protein Assay Kit (Bio-Rad Laboratories, Hercules, CA, USA) and the MultiSkan GO spectrophotometer (Thermo Fisher Scientific Inc., Waltham, MA, USA). The proteins of the samples were separated via SDS-polyacrylamide gel electrophoresis using the Mini Protean Tetra Cell equipment (Bio-Rad Laboratories, Hercules, CA, USA). After the separation, the proteins were blotted onto nitrocellulose membranes (Amersham Biosciences, GE Healthcare, Amersham, UK). The membranes were blocked for 1 h with TBST buffer containing 5% (*w*/*v*) non-fat dry milk at room temperature. The following primary antibodies were used in the experiments: anti-TfR1 IgG (1:1000, 1 h, room temperature, Thermo Fisher Scientific Inc., Waltham, MA, USA); anti-DMT-1 IgG (1:1000, 1 h, room temperature, Thermo Fisher Scientific Inc., Waltham, MA, USA) anti-FP IgG (1:1000, 1 h, room temperature, Bio-Techne, Minneapolis, MN, USA); anti-FECH IgG (1:1000, 1 h, room temperature, Thermo Fisher Scientific Inc., Waltham, MA, USA); anti-HO-1 IgG (1:1000, overnight, 4 °C, Cell Signaling Technology Europe, Leiden, The Netherlands); anti-FTH (1:1000, overnight, 4 °C, Cell Signaling Technology Europe, Leiden, The Netherlands), anti-Mfrn-1 IgG (1:1000, 1 h, room temperature, Thermo Fisher Scientific Inc., Waltham, MA, USA) and anti-Mfrn-2 IgG (1:1000, 1 h, room temperature, Thermo Fisher Scientific Inc., Waltham, MA, USA). The glyceraldehyde 3-phosphate dehydrogenase (anti-GAPDH IgG, 1:3000; Merck Life Science Kft., Budapest, Hungary, cat.no.: G9545) was used as a loading control. For the secondary antibody, goat anti-rabbit IgG (H + L) HRP conjugated IgG was used (1:3000; Merck Life Science Kft., Budapest, Hungary) for 1 h at room temperature. The UVItec Alliance Q9 Advanced Imaging System (UVItec Cam-bridge Ltd., Cambridge, UK) was used for visualization. The WesternBright ECL chemiluminescent substrate (Advansta Inc., San Jose, CA, USA) was used for the development of the blots. The optical densities were determined using ImageJ software version IJ153 [54]. The protein levels were expressed as a percentage of the target protein/GAPDH ratio. Western blots shown in the figures were representative of three independent experiments. We carried out negative controls for each primary antibody using only the secondary antibody to prove the specificity of the reactions. These blots can be seen in Appendix A.

### 4.4. Enzyme-Linked Immunosorbent Assay (ELISA)

The supernatants of the treated and control HEC-1A cells were collected and stored at −80 °C until the ELISA measurements. The secreted haptoglobin concentration was determined with a Human Haptoglobin Quantikine ELISA kit (Bio-Techne, Minneapolis, MN, USA) according to the protocol of the manufacturer. The measurements were performed in triplicate in three independent experiments.

### 4.5. Total Iron Measurements

The HEC-1A cells were cultured in 6 cm-wide cell culture dishes (Biologix Europe, Hallbergmoos, Germany) using 10^6^ cells/dish. After the treatments, the cells were harvested via centrifugation and were lysed in 50 mM NaOH for 2 h at room temperature with gentle shaking. The samples were neutralized using 10 mM HCl, and were then mixed with acidic KMnO_4_ and incubated at 60 °C for 2 h for the liberation of protein-bound iron. After cooling the mixtures to room temperature, reduction and complex formation were carried out using 6.5 mM neocuproine, 1 M ascorbic acid, 2.5 M ammonium-acetate and 6.5 mM ferrozine at room temperature for 30 min. During the reaction, ferrozine forms a purple complex with the ferrous iron ions, which is detectable at 550 nm. The absorbance of the samples was measured using a MultiSkan GO spectrophotometer (Thermo Fisher Scientific Inc., Waltham, MA, USA). The concentration of the samples was determined against the FeCl_3_ standard curve and was expressed as mM iron/mg protein after the normalization to the protein content of each sample. The measurements were carried out in triplicate in three independent experiments. All reagents were purchased from Merck Life Science (Merck Life Science Kft., Budapest, Hungary).

### 4.6. Determination of the Heme Concentration

The HEC-1A cells were cultured in 6-well plates using 5 × 10^5^ cells/well for the heme measurements. After the treatments, the cells were collected and lysed in distilled water using gentle sonication. The heme concentration measurements were carried out using a Heme Assay Kit (Merck Life Science Kft., Budapest, Hungary). Each sample was mixed with a four-fold Heme Reagent and incubated for 5 min at room temperature. The absorbance was measured at 400 nm, and the results were expressed as μM. The measurements were carried out in triplicate in three independent experiments.

### 4.7. Data Analysis

Data are shown as the mean ± standard deviation (SD). Statistical analysis was performed using SPSS software version 24.0 (IBM Corporation, Armonk, NY, USA). Statistical significance was determined by two-way ANOVA followed by Scheffe’s post hoc test. The results were considered statistically significant if the *p*-value was <0.05.

## 5. Conclusions

According to the observations, FKN can improve the deteriorating effect of iron deficiency in HEC-1A endometrium cells by modifying iron uptake, release and storage. FKN may also contribute to maintaining mitochondrial iron homeostasis. FKN may cause iron redistribution and modifies iron utilization of the endometrium cells, which may contribute to the development of receptivity and/or ensure iron transport towards the embryo at the early stage of pregnancy.

## Figures and Tables

**Figure 1 ijms-24-09917-f001:**
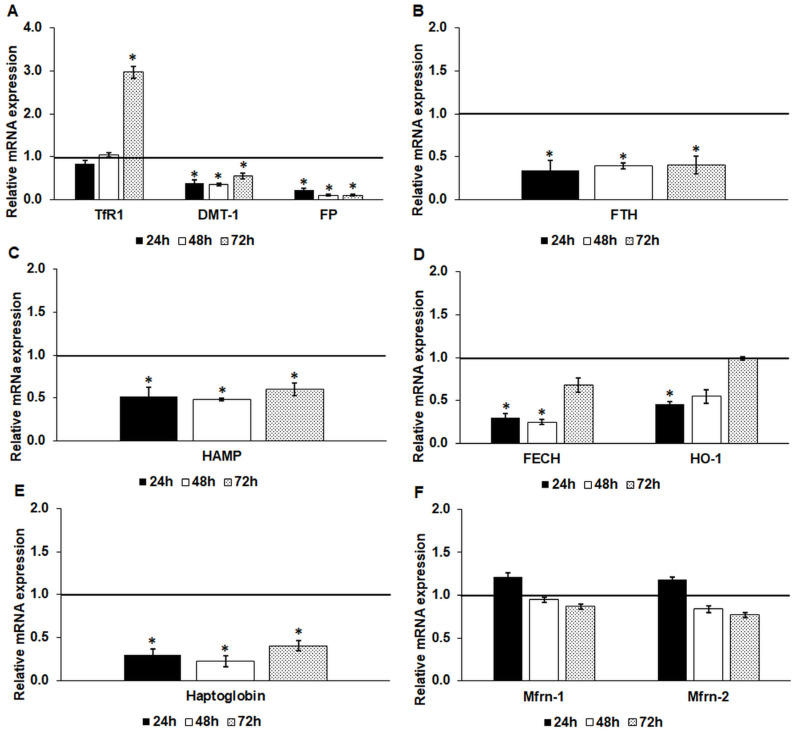
Real-time PCR analysis of the mRNA expression of the iron metabolism-related genes after 24 h-, 48 h- and 72 h-long DFO treatments in a serum-free culture medium. Real-time PCR was performed using a SYBR green protocol. GAPDH was used as a housekeeping gene to normalize the gene expression levels. The untreated cells were used as a control in the experiment. (**A**) mRNA expression levels of iron transporters TfR1, DMT-1 and FP. (**B**) The mRNA expression level of the cytoplasmic iron storage protein FTH. (**C**) Relative mRNA level of the HAMP gene. (**D**) Expression levels of FECH and HO-1 involved in heme metabolism. (**E**) The mRNA expression level of the hemoglobin transporter haptoglobin. (**F**) Relative mRNA levels of the mitochondrial iron transporters Mfrn-1 and Mfrn-2. The relative expression level of the target genes of the control was regarded as 1, which is represented by the horizontal line. The columns represent the mean ± SD of three independent experiments (*n* = 3). The analysis was carried out in triplicate/sample in each experiment. The asterisk shows *p* < 0.05 compared to the control. Abbreviations: DFO-desferrioxamine; TfR1-transferrin receptor 1; DMT-1-divalent metal transporter-1; FP-ferroportin; HAMP-hepcidin gene; FECH-ferrochelatase; HO-1-heme oxygenase-1; Mfrn-1-mitoferrin-1; Mfrn-2-mitoferrin-2.

**Figure 2 ijms-24-09917-f002:**
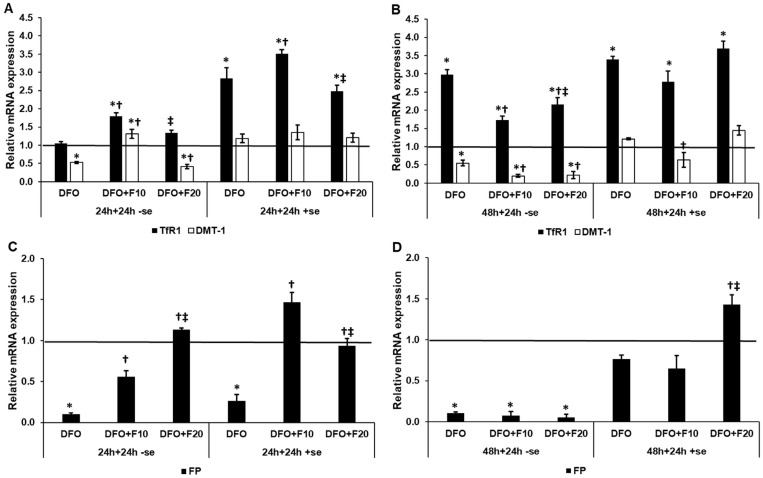
Real-time PCR analysis of the mRNA expression of the iron transporter TfR1, DMT−1 and FP in HEC-1A cells after 24 and 48 h DFO treatments in a serum-free environment followed by 24 h FKN addition in a serum-free or serum-supplemented culture medium. Real-time PCR was performed using a SYBR green protocol. GAPDH was used as a housekeeping gene for the normalization of the expression levels. The untreated cells were used as a control in the experiments (**A**) mRNA expression levels of TfR1 and DMT−1 after 24 h DFO treatment in a serum-free environment followed by 24 h FKN addition in a serum-free or serum-supplemented culture medium. (**B**) Expression levels of TfR1 and DMT−1 after 48 h DFO treatment in a serum-free environment followed by 24 h FKN addition in a serum-free or serum-supplemented culture medium. (**C**) Relative mRNA levels of FP after 24 h DFO treatment in a serum-free environment followed by 24 h FKN addition in a serum-free or serum-supplemented culture medium. (**D**) Relative mRNA levels of FP after 48 h DFO treatment in a serum-free environment followed by 24 h FKN addition in a serum-free or serum-supplemented culture medium. The relative expression level of the target genes of the control was considered as 1, which is represented by the horizontal line. The columns represent the mean ± SD of three independent experiments (*n* = 3). The analysis was carried out in triplicate/sample in each experiment. The asterisk shows *p* < 0.05 compared to the control. The cross signs *p* < 0.05 compared to the DFO treatments. The double cross means *p* < 0.05 compared to the F10 treatment. Abbreviations: TfR1-transferrin receptor 1; DMT−1-divalent metal transporter-1; FP-ferroportin; DFO-desferrioxamine; F10-fractalkine 10 ng/mL, F20-fractalkine 20 ng/mL; -se-serum-free; +se-serum supplemented.

**Figure 3 ijms-24-09917-f003:**
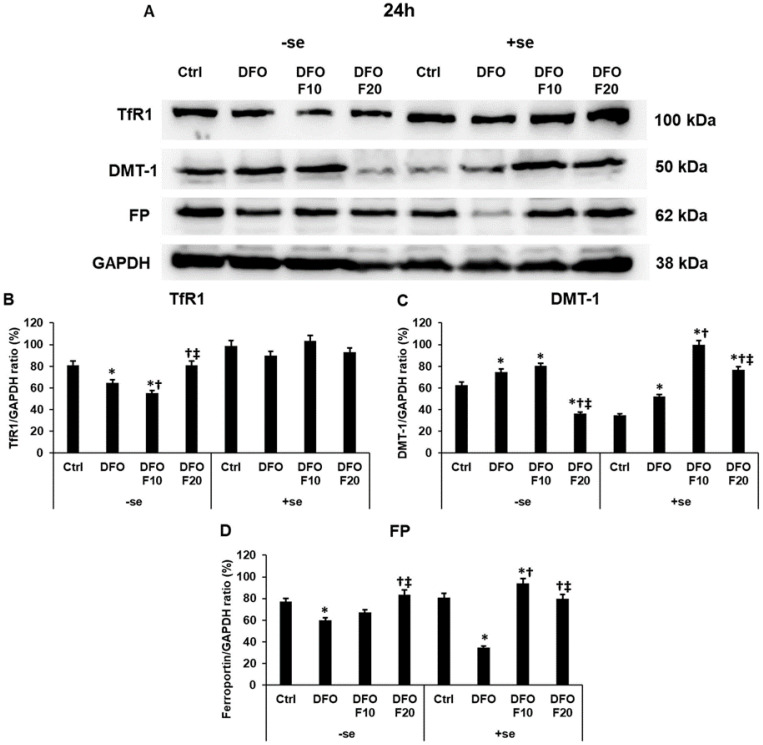
Western blot analysis of TfR1, DMT−1 and FP in HEC-1A cells after 24 h DFO treatment in a serum-free environment followed by 24 h FKN addition in a serum-free or serum-supplemented culture medium. After the treatments, the cells were collected and lysed, and then the same amount of protein from each sample was separated by 10% SDS-PAGE. After blotting, the membranes were probed with anti-TfR1, anti-DMT−1 or anti-FP according to the manufacturer’s instructions. The experiments were repeated three times. GAPDH was used as the loading control. Analysis of the WBs was performed using ImageJ Software IJ153. (**A**) Representative Western blots of TfR1, DMT−1 and FP. (**B**–**D**) Optical density analyses of the target proteins after 24 h DFO treatment followed by 24 h FKN addition in a serum-free or serum-supplemented culture medium. The blots were cropped according to the molecular weight of the target protein. The original blots can be found in the Appendix A. The columns represent the mean ± SD of three independent experiments (*n* = 3). The asterisk shows *p* < 0.05 compared to the control. The cross shows *p* < 0.05 compared to the DFO treatments. The double cross signs *p* < 0.05 compared to the F10 treatment. Abbreviations: TfR1-transferrin receptor 1; DMT−1-divalent metal transporter-1; FP-ferroportin; DFO-desferrioxamine; FKN-fractalkine; F10-fractalkine 10 ng/mL, F20-fractalkine 20 ng/mL; -se-serum-free; +se-serum supplemented.

**Figure 4 ijms-24-09917-f004:**
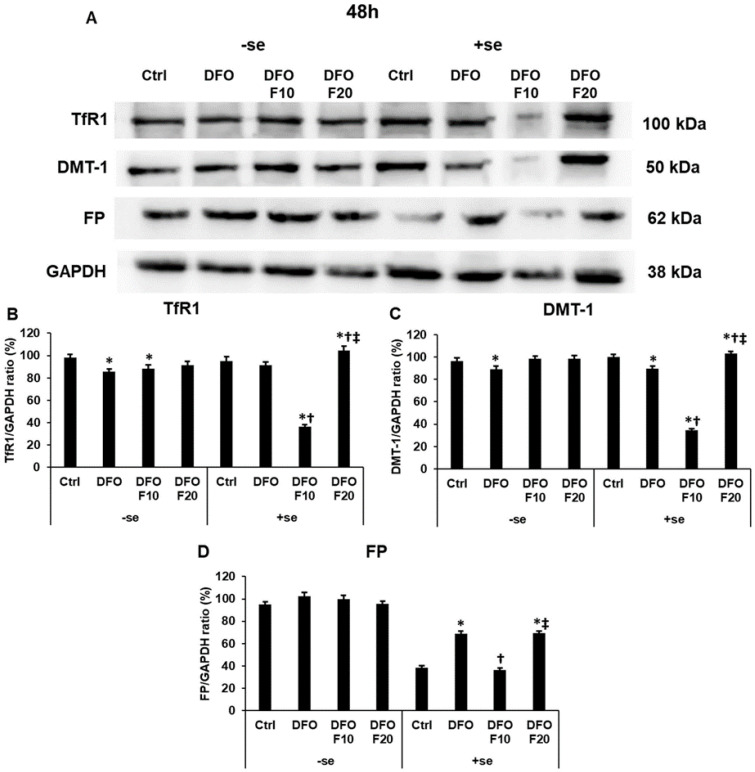
Western blot analysis of TfR1, DMT−1 and FP in HEC-1A cells after 48 h DFO treatment in a serum-free environment followed by 24 h FKN addition in a serum-free or serum-supplemented culture medium. After the treatments, the cells were collected and lysed, and then the same amount of protein from each sample was separated by 10% SDS-PAGE. After blotting, the membranes were probed with anti-TfR1, anti-DMT−1 or anti-FP according to the manufacturer’s instructions. The experiments were repeated three times. GAPDH was used as the loading control. Analysis of the WBs was performed using ImageJ Software IJ153. (**A**) Representative Western blots of TfR1, DMT−1 and FP. (**B**–**D**) Optical density analyses of the target proteins after 48 h DFO treatment followed by 24 h FKN addition in a serum-free or serum-supplemented culture medium. The blots were cropped according to the molecular weight of the target protein. The original blots can be found in the Appendix A. The columns represent the mean ± SD of three independent experiments (*n* = 3). The asterisk shows *p* < 0.05 compared to the control. The cross signs *p* < 0.05 compared to the DFO treatments. The double cross means *p* < 0.05 compared to the F10 treatment. Abbreviations: TfR1-transferrin receptor 1; DMT-1-divalent metal transporter-1; FP-ferroportin; DFO-desferrioxamine; FKN-fractalkine; F10-fractalkine 10 ng/mL, F20-fractalkine 20 ng/mL; -se-serum-free; +se-serum supplemented.

**Figure 5 ijms-24-09917-f005:**
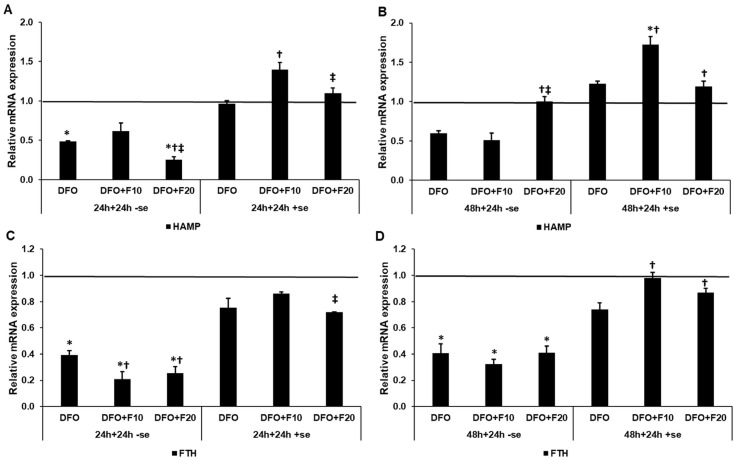
Real time PCR analysis of the mRNA expression of HAMP and FTH in HEC-1A cells after 24 and 48 h DFO treatments in a serum-free environment followed by 24 h FKN addition in a serum-free or serum-supplemented culture medium. Real-time PCR was performed using a SYBR green protocol. GAPDH was used as a housekeeping gene to normalize the gene expression levels. The untreated cells were used as a control in the experiments. (**A**) mRNA expression levels of HAMP after 24 h DFO treatment in a serum-free environment followed by 24 h FKN addition in a serum-free or serum-supplemented culture medium. (**B**) Expression levels of HAMP after 48 h DFO treatment in a serum-free environment followed by 24 h FKN addition in a serum-free or serum-supplemented culture medium. (**C**) Relative mRNA levels of FTH after 24 h DFO treatment in a serum-free environment followed by 24 h FKN addition in a serum-free or serum-supplemented culture medium. (**D**) Relative mRNA levels of FTH after 48 h DFO treatment in a serum-free environment followed by 24 h FKN addition in a serum-free or serum-supplemented culture medium. The relative expression levels of the target genes of the control were regarded as 1, which is represented by the horizontal line. The columns represent the mean ± SD of three independent experiments (*n* = 3). The analysis was carried out in triplicate/sample in each experiment. The asterisk shows *p* < 0.05 compared to the control. The cross signs *p* < 0.05 compared to the DFO treatments. The double cross means *p* < 0.05 compared to the F10 treatment. Abbreviations: HAMP-hepcidin gene; FTH-ferritin heavy chain; DFO-desferrioxamine; FKN-fractalkine; F10-fractalkine 10 ng/mL, F20-fractalkine 20 ng/mL; -se-serum-free; +se-serum supplemented.

**Figure 6 ijms-24-09917-f006:**
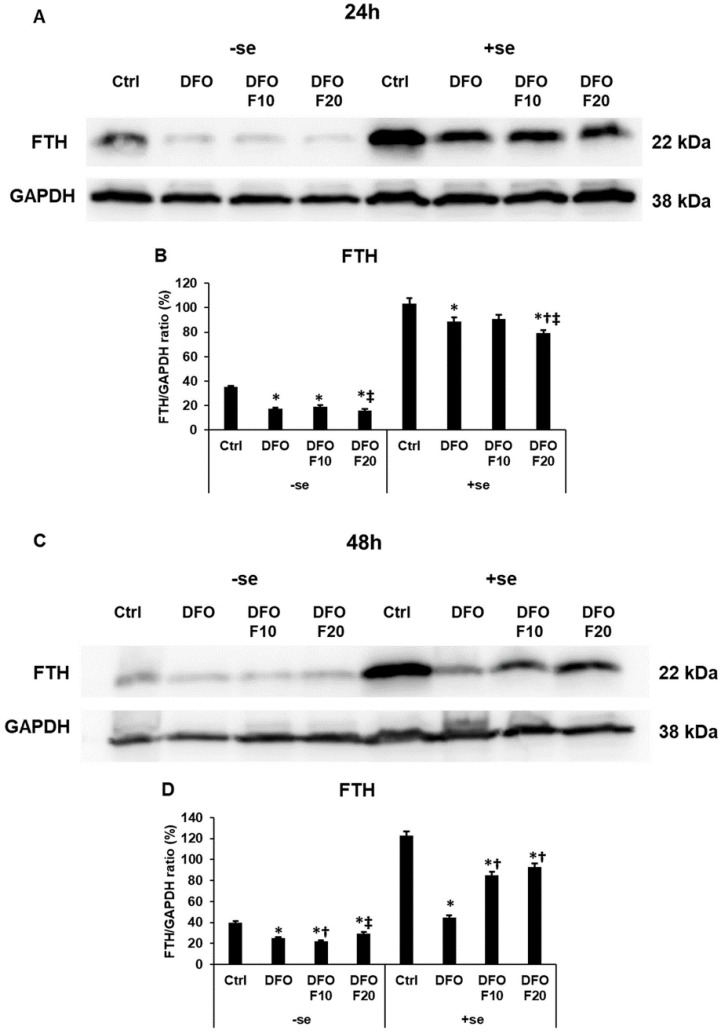
Western blot analysis of cytosolic iron storage protein FTH in HEC-1A cells after 24 h (**A**,**B**) and 48 h (**C**,**D**) DFO treatments in a serum-free environment followed by 24 h FKN addition in a serum-free or serum-supplemented culture medium. After the treatments, the cells were collected and lysed, and then the same amount of protein from each sample was separated by 14% SDS-PAGE. After blotting, the membranes were probed with anti-FTH primary antibodies according to the manufacturer’s instructions. The experiments were repeated three times. GAPDH was used as the loading control. (**A**) Representative Western blot of FTH after 24 h DFO treatment in a serum-free environment followed by 24 h FKN addition in a serum-free or serum-supplemented culture medium. (**B**) Optical density analysis of FTH after 24 h DFO treatment followed by 24 h FKN addition in a serum-free or serum-supplemented culture medium. (**C**) Representative Western blot of FTH after 48 h DFO treatment in a serum-free environment followed by 24 h FKN addition in a serum-free or serum-supplemented culture medium. (**D**) Optical density analysis of FTH after 48 h DFO treatment followed by 24 h FKN addition in a serum-free or serum-supplemented culture medium. The blots were cropped according to the molecular weight of the target protein. The original blots can be found in the Appendix A. The columns represent the mean ± SD of three independent experiments (*n* = 3). The asterisk shows *p* < 0.05 compared to the control. The cross signs *p* < 0.05 compared to the DFO treatments. The double cross means *p* < 0.05 compared to the F10 treatment. Abbreviations: FTH-ferritin heavy chain; DFO-desferrioxamine; FKN-fractalkine; F10-fractalkine 10 ng/mL, F20-fractalkine 20 ng/mL; -se-serum-free; +se-serum supplemented.

**Figure 7 ijms-24-09917-f007:**
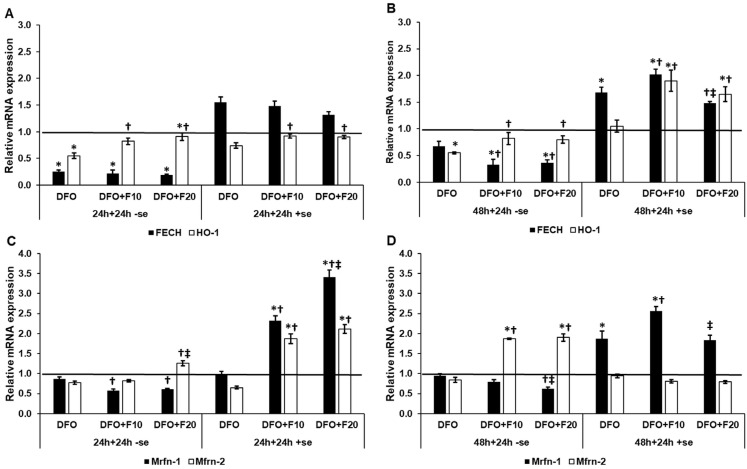
Real-time PCR analysis of the mRNA expression of the genes related to mitochondrial iron utilization in HEC-1A cells after 24 and 48 h DFO treatments in a serum-free environment followed by 24 h FKN addition in a serum-free or serum-supplemented culture medium. Real-time PCR was performed using an SYBR green protocol. GAPDH was used as a housekeeping gene to normalize the gene expression levels. The untreated cells were used as a control in the experiments. (**A**) mRNA expression levels of FECH and HO-1 after 24 h DFO treatment in a serum-free environment followed by 24 h FKN addition in a serum-free or serum-supplemented culture medium. (**B**) Expression levels of FECH and HO-1 after 48 h DFO treatment in a serum-free environment followed by 24 h FKN addition in a serum-free or serum-supplemented culture medium. (**C**) Relative mRNA levels of Mfrn-1 and Mfrn-2 after 24 h DFO treatment in a serum-free environment followed by 24 h FKN addition in a serum-free or serum-supplemented culture medium. (**D**) Relative mRNA levels of Mfrn-1 and Mfrn-2 after 48 h DFO treatment in a serum-free environment followed by 24 h FKN addition in a serum-free or serum-supplemented culture medium. The relative expression level of the target genes of the control was regarded as 1, which is presented by the horizontal line. The columns represent the mean ± SD of three independent experiments (*n* = 3). The analysis was carried out in triplicate/sample in each experiment. The asterisk shows *p* < 0.05 compared to the control. The cross signs *p* < 0.05 compared to the DFO treatments. The double cross means *p* < 0.05 compared to the F10 treatment. Abbreviations: FECH-ferrochelatase; HO-1-heme oxigenase-1; Mfrn-1-mitoferrin-1; Mfrn-2-mitoferrin-2; DFO-desferrioxamine; FKN-fractalkine; F10-fractalkine 10 ng/mL, F20-fractalkine 20 ng/mL; -se-serum-free; +se-serum supplemented.

**Figure 8 ijms-24-09917-f008:**
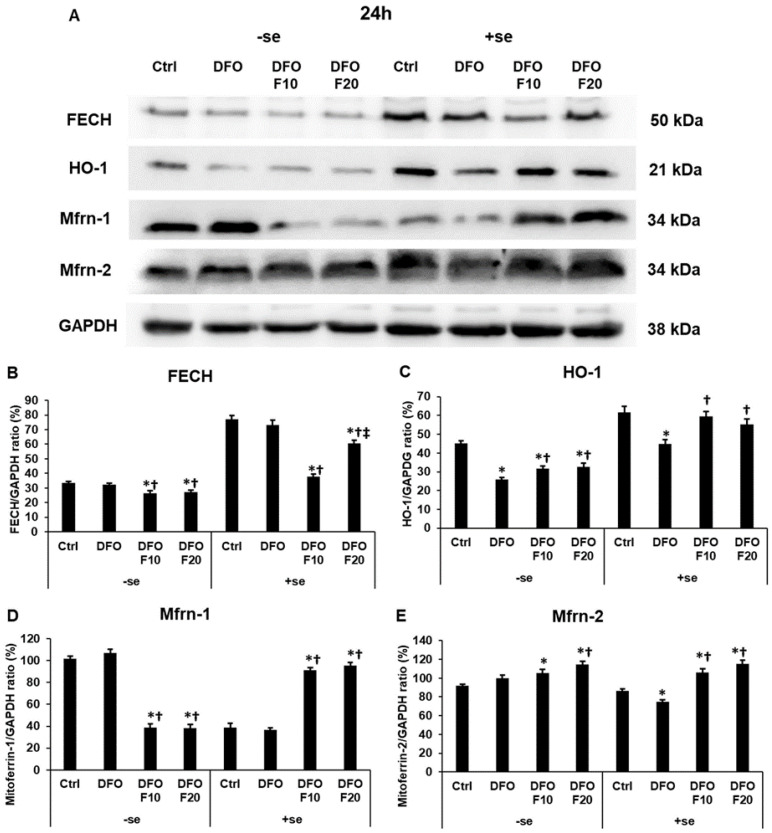
Determination of the expression levels of proteins related to mitochondrial iron utilization after 24 h DFO treatment in a serum-free environment followed by 24 h FKN addition in a serum-free or serum-supplemented culture medium. The HEC-1A cells were collected after the treatments, then cells were lysed, and the same amount of protein (10 μg) from each sample was separated by SDS-PAGE using 12 or 14% polyacrylamide gels. After blotting to nitrocellulose membranes, the membranes were probed with anti-FECH, anti-HO-1, anti-Mfrn-1, and anti-Mfrn-2 according to the manufacturer’s instructions. The experiments were repeated three times. GAPDH was used as the loading control. Analysis of the WBs was performed using ImageJ Software IJ153. (**A**) Representative Western blots of FECH, HO-1, Mfrn-1 and Mfrn-2. (**B**–**E**) Optical density analyses of FECH, HO-1, Mfrn-1 and Mfrn-2 in HEC-1A cells after 24 h DFO treatment followed by 24 h FKN addition in a serum-free or serum-supplemented culture medium. The blots were cropped according to the molecular weight of the target protein. The original blots can be found in the Appendix A. The columns represent the mean ± SD of three independent experiments (*n* = 3). The asterisk shows *p* < 0.05 compared to the control. The cross signs *p* < 0.05 compared to the DFO treatments. The double cross means *p* < 0.05 compared to the F10 treatment. Abbreviations: FECH-ferrochelatase; HO-1-heme oxigenase-1; Mfrn-1-mitoferrin-1; Mfrn-2-mitoferrin-2; DFO-desferrioxamine; FKN- fractalkine; F10-fractalkine 10 ng/mL, F20-fractalkine 20 ng/mL; -se-serum-free; +se-serum supplemented.

**Figure 9 ijms-24-09917-f009:**
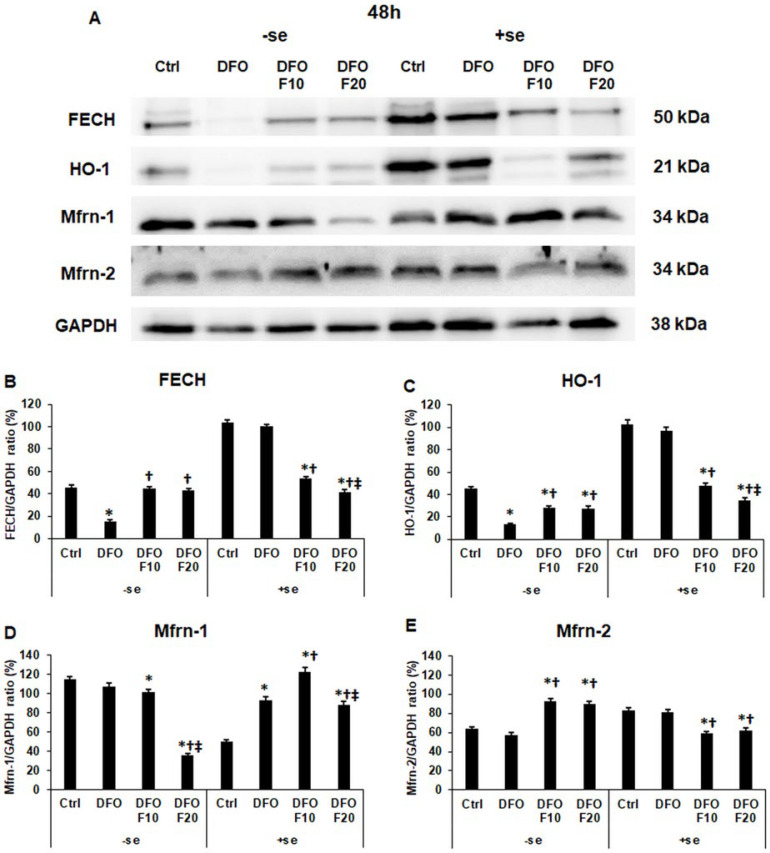
Determination of the expression levels of proteins related to mitochondrial iron utilization after 48 h DFO treatment in a serum-free environment followed by 24 h FKN addition in a serum-free or serum-supplemented culture medium. The HEC-1A cells were collected after the treatments, then cells were lysed, and the same amount of protein (10 μg) from each sample was separated by SDS-PAGE using 12 or 14% polyacrylamide gels. After blotting to nitrocellulose membranes, the membranes were probed with anti-FECH, anti-HO-1, anti-Mfrn-1, and anti-Mfrn-2 according to the manufacturer’s instructions. The experiments were repeated three times. GAPDH was used as the loading control. Analysis of the WBs was performed using ImageJ Software IJ153. (**A**) Representative Western blots of FECH, HO-1, Mfrn-1 and Mfrn-2. (**B**–**E**) Optical density analyses of FECH, HO-1, Mfrn-1 and Mfrn-2 in HEC-1A cells after 48 h DFO treatment followed by 24 h FKN addition in serum-free or serum-supplemented culture media. The blots were cropped according to the molecular weight of the target protein. The original blots can be found in the Appendix A. The columns represent the mean ± SD of three independent experiments (*n* = 3). The asterisk shows *p* < 0.05 compared to the control. The cross signs *p* < 0.05 compared to the DFO treatments. The double cross means *p* < 0.05 compared to the F10 treatment. Abbreviations: FECH-ferrochelatase; HO-1-heme oxigenase-1; Mfrn-1-mitoferrin-1; Mfrn-2-mitoferrin-2; DFO-desferrioxamine; FKN- fractalkine; F10-fractalkine 10 ng/mL, F20-fractalkine 20 ng/mL; -se-serum-free; +se-serum supplemented.

**Figure 10 ijms-24-09917-f010:**
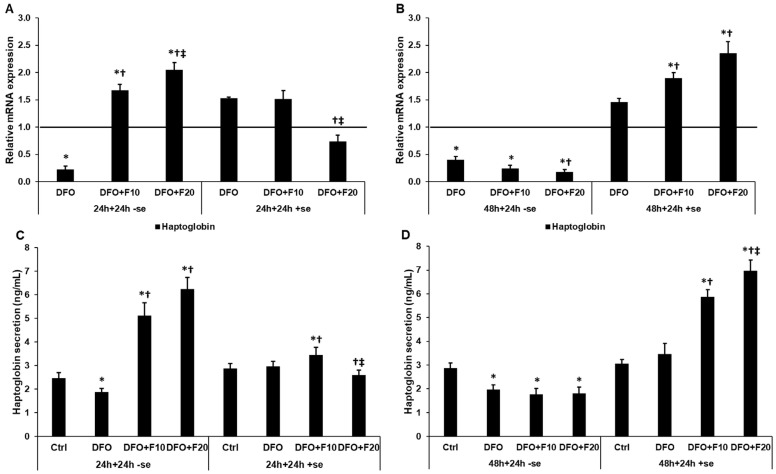
mRNA expression analysis and ELISA measurements of haptoglobin after 24 h and 48 h-long DFO treatments in a serum-free environment followed by 24 h FKN supplementation in serum-free and serum-containing culture media. Real-time PCR was performed using an SYBR green protocol. GAPDH was used as a housekeeping gene for the normalization of the expression levels. The untreated cells were used as a control in the experiment. The relative expression level of the haptoglobin gene of the control was regarded as 1, which is presented by the horizontal line. ELISA measurements were carried out using human haptoglobin ELISA kit following the manufacturer’s protocols. (**A**) mRNA levels of haptoglobin after 24 h DFO treatment followed by 24 h FKN supplementation in a serum-free and serum-containing culture media. (**B**) mRNA levels of haptoglobin after 48 h DFO treatment followed by 24 h FKN supplementation in serum-free and serum-containing environments. (**C**) Secreted haptoglobin concentration after 24 h DFO treatment followed by 24 h FKN supplementation in a serum-free and serum-containing culture media. (**D**) Secreted haptoglobin concentration after 48 h DFO treatment followed by 24 h FKN supplementation in a serum-free and a serum-containing culture media. The columns represent the mean ± SD of three independent experiments (*n* = 3). The determinations were carried out in triplicate/sample in each experiment. The asterisk shows *p* < 0.05 compared to the control. The cross signs *p* < 0.05 compared to the DFO treatments. The double cross means *p* < 0.05 compared to the F10 treatment. Abbreviations: DFO-desferrioxamine; FKN-fractalkine; F10-fractalkine 10 ng/mL, F20-fractalkine 20 ng/mL; -se-serum-free; +se-serum supplemented.

**Figure 11 ijms-24-09917-f011:**
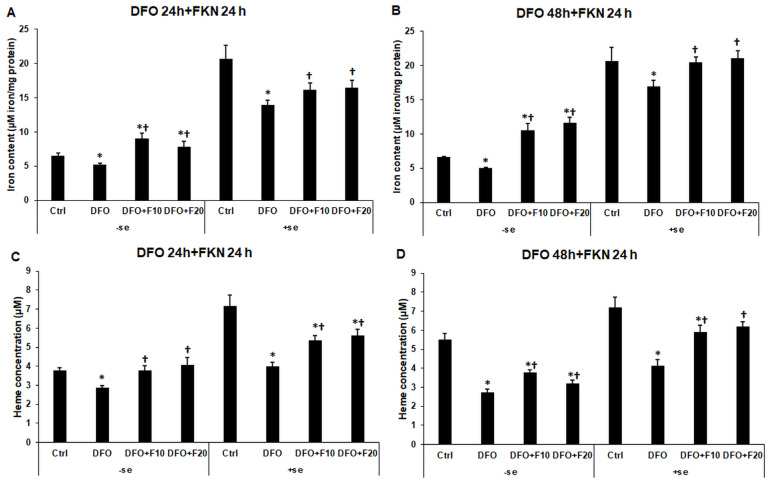
Determination of the total intracellular iron content and heme concentration in HEC-1A cells after 24 h and 48 h-long DFO treatments in a serum-free environment followed by 24 h FKN supplementation in serum-free and serum-containing culture media. Total iron measurements were performed using a colorimetric ferrozine-based method. The results were expressed as μM iron/mg protein. Heme concentration measurements were carried out by a Heme Assay Kit. The measurements were carried out in triplicate in three independent experiments. (**A**) The total iron content of the HEC-1A cells after 24 h DFO treatment in a serum-free environment followed by 24 h FKN supplementation in serum-free and serum-containing culture media. (**B**) The total iron content of the HEC-1A cells after 48 h DFO treatment in a serum-free environment followed by 24 h FKN supplementation in serum-free and serum-containing culture media. (**C**) Heme concentration of the HEC-1A cells after 24 h DFO treatment in a serum-free environment followed by 24 h FKN supplementation in serum-free and serum-containing culture media. (**D**) Heme concentration of the HEC-1A cells after 48 h DFO treatment in a serum-free environment followed by 24 h FKN supplementation in serum-free and serum-containing culture media. The columns represent the mean ± SD of three independent experiments (*n* = 3). The asterisk shows *p* < 0.05 compared to the control. The cross signs *p* < 0.05 compared to the DFO treatments. Abbreviations: DFO-desferrioxamine; FKN-fractalkine; F10-fractalkine 10 ng/mL, F20-fractalkine 20 ng/mL; -se-serum-free; +se-serum supplemented.

**Figure 12 ijms-24-09917-f012:**
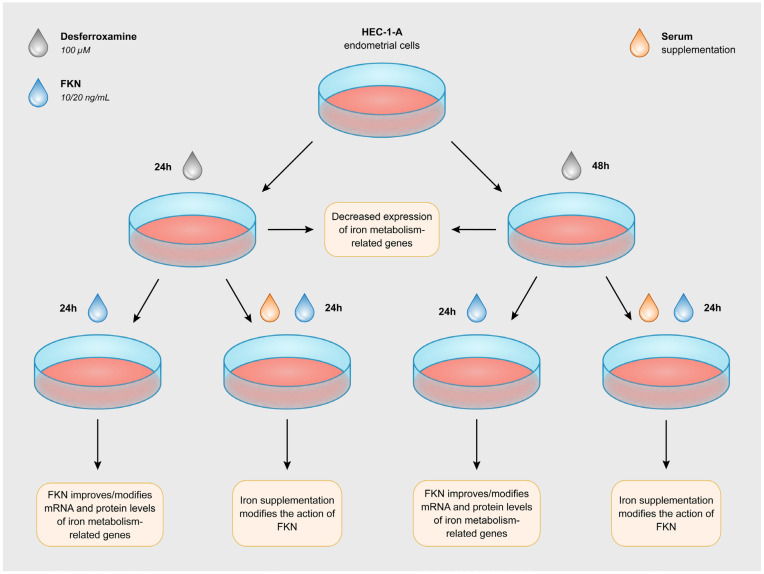
Experimental setup. iron deficiency was generated for 24 h and 48 h using 100 μM DFO. Then, the cell cultures were separated into two groups. The first group was supplemented with recombinant human fractalkine at 10 and 20 ng/mL concentrations in a serum-free environment for 24 h. The second group was treated with 10 and 20 ng/mL fractalkine together with serum supplementation for 24 h. In the presence of serum, iron was again available for the cells. In each experiment, untreated cells were used as controls.

**Table 1 ijms-24-09917-t001:** Real-time primer list.

Primer	Sequence 5′ → 3′
HAMP forward	CAGCTGGATGCCCATGTT
HAMP reverse	TGCAGCACATCCCACATC
GAPDH forward	TGTTCCAATATGATTCCACCC
GAPDH reverse	CCACTTGATTTTGGAGGGAT
TfR1 forward	CATGTGGAGATGAAACTTGC
TfR1 reverse	TCCCATAGCAGATACTTCCA
DMT-1 forward	GTGGTTACTGGGCTGCATCT
DMT-1 reverse	CCCACAGAGGAATTCTTCCT
FP forward	AAAGGAGGCTGTTTCCATAG
FP reverse	TTCCTTCTCTACCTTGGTCA
FTH forward	GAGGTGGCCGAATCTTCCTTC
FTH reverse	TCAGTGGCCAGTTTGTGCAG
FECH forward	GATGAATTGTCCCCCAACAC
FECH reverse	AGCCCTTTCTAGGCCATCTC
HO-1 forward	ACCCATGACACCAAGGACCA
HO-1 reverse	ATGCCTGCATTCACATGGCA
Haptoglobin forward	CAAGAAGACACCTGCTATGG
Haptoglobin reverse	GTCACCTTCACATACACACC
Mfrn-1 forward	TTAAATGACGTTTTCCACCAC
Mfrn-1 reverse	TTCTGCTGGATTCATTACCG
Mfrn-2 forward	ATAGCCATATTGCCAATGGTG
Mfrn-2 reverse	CCGGTGGTATGGTGAGTTGTA

## Data Availability

The data underlying this article are available in the article and its online Appendix A.

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
