# Peer review of "The Role of Fractalkine in the Regulation of Endometrial Iron Metabolism in Iron Deficiency"

_ijms, 2023, doi:10.3390/ijms24129917_

Round 1

Reviewer 1 Report

VEry nice English.  This is an in vitro study using adenocarcinoma cells to imply in vivo endometrial function concerning the biology of iron in the endometrium of humans during the peri-implantation period of pregnancy.  As such is has the inherent weakness of being entirely in vitro, but has the strength of providing a substantial amount of mechanistic data.  I for one am not bothered by the in vitro nature of the study and think there is merit to the insights provided by the mechanistic results.  I do have three concerns.

1) The authors do not show the values for their controls in the pcr data.  These need to be shown in bar form in the bar graphs.  Indicating a significant difference from a value you do not even show in the graph makes it difficult to assess what is really going on.

2) The authors need to include IgG controls for the Western blots.  J Histochem Cytochem 2104; 62:693 highlights the importance of such controls for immunohistochemistry and Westerns require the same.

3) This is an adenocarcinoma cell line.  The glands in the endometrium are certainly important, but so are all the other cell types in the endometrium.  It is disappointing that the authors in their introduction and discussion treat the endometrium like it is generically one cell type and never split out what might be going on physiologically in the different cell types within the endometrium.  I think the Discussion could be greatly improved by taking into account some histological ramifications to the results.

Author Response

Answers for Reviewer 1.

Very nice English. This is an in vitro study using adenocarcinoma cells to imply in vivo endometrial function concerning the biology of iron in the endometrium of humans during the peri-implantation period of pregnancy.  As such is has the inherent weakness of being entirely in vitro but has the strength of providing a substantial amount of mechanistic data.  I for one am not bothered by the in vitro nature of the study and think there is merit to the insights provided by the mechanistic results.  I do have three concerns.

1) The authors do not show the values for their controls in the pcr data.  These need to be shown in bar form in the bar graphs.  Indicating a significant difference from a value you do not even show in the graph makes it difficult to assess what is really going on.

Thank you for the comment. The mRNA expression level of the control cells was regarded as 1 in each real-time PCR experiment. According to the high number of examined target genes, we should add four more columns to each figure (Fig. 2,5,7, and 10), making the figures smaller and more crowded. Instead of adding columns, a horizontal line was added to each graph presenting mRNA expression levels at value 1, making the columns more comparable to the control.

2) The authors need to include IgG controls for the Western blots.  J Histochem Cytochem 2104; 62:693 highlights the importance of such controls for immunohistochemistry and Westerns require the same.

Thank you for the suggestion. Western blots with the primary antibody utilized for the immunohistochemistry examination can be normally used as positive controls of the immunohistochemistry experiments for the verification of the antibody specificity. The determination of the specificity of the primary antibody in the case of immunohistochemistry or -cytochemistry needs proper positive and negative controls, since in these techniques we cannot see whether the signal is coming from one protein (target protein) or non-specific protein-antibody interactions (false positive signal). In the case of WB, it is simpler, if the primary antibody is not specific for the target protein, more than one strong protein band can be observed with different molecular weights. If the secondary antibody “sticks” to the membrane at non-specific sites, a strong background appears on the blots during detection. We usually use negative controls for the WB-s, which show that without the primary antibody no signal is generated (no non-specific binding of the secondary antibody), and a loading control protein exhibiting high-level and constitutive expression in the examined cells, which shows the amount of proteins used for the separation. In examining a recombinant protein, it is recommended to use an endogenous control to see whether the antibody recognizes the epitope on the recombinant protein.

We have carried out negative controls for the WB-s, that now can be seen in the supplementary material.

3) This is an adenocarcinoma cell line.  The glands in the endometrium are certainly important, but so are all the other cell types in the endometrium.  It is disappointing that the authors in their introduction and discussion treat the endometrium like it is generically one cell type and never split out what might be going on physiologically in the different cell types within the endometrium.  I think the Discussion could be greatly improved by taking into account some histological ramifications to the results.

Thank you for the advice. The Discussion section was supplemented with the requested parts.

All changes can be followed by the track change mode.

Reviewer 2 Report

Dear authors,

I would like to recommend you further studies at the cellular / ultrastructural levels (e.g. cytology , TEM, SEM, AFM,  AFM), in relationships to mitochondrial activation ic your in vitro experimmentsl conditions, which are interesting modelling of iron methabolism in non-erythroid cells.

Additionally, I could point out that paper reviewed is very well prepared from the molecular biology point of view.   But from the place of cell biology,  when cell proliferative factors as nuclear gactor kappa B,  and cell organelles like mitochondria are topics and objects of investigation, it will be fruitful to apply simultaneously cytological and ultrastructural methods (Ali  M., et al., Cells 2022; Zvetkova E., and Yaroslav Jelinek, 1989). The paper of M.Ali et al., has been cited in References of your manuscript.   By this way, cell line examined in the in vitro model of authors, will be very well characterized. The cellular language will rest unknown, in the absence of morphological studies (my own opinion).   On the other hand, the authors are largely known molecular biologists, (please see References), and their experimental results, hypotheses and opinions are undoubtedly accepted by me and by the international scientific society.

Author Response

Answers for Reviewer 2.

I would like to recommend you further studies at the cellular / ultrastructural levels (e.g. cytology , TEM, SEM, AFM,  AFM), in relationships to mitochondrial activation ic your in vitro experimental conditions, which are interesting modelling of iron metabolism in non-erythroid cells.

Additionally, I could point out that paper reviewed is very well prepared from the molecular biology point of view.   But from the place of cell biology, when cell proliferative factors as nuclear gactor kappa B, and cell organelles like mitochondria are topics and objects of investigation, it will be fruitful to apply simultaneously cytological and ultrastructural methods (Ali  M., et al., Cells 2022; Zvetkova E., and Yaroslav Jelinek, 1989). The paper of M.Ali et al., has been cited in References of your manuscript.   By this way, cell line examined in the in vitro model of authors, will be very well characterized. The cellular language will rest unknown, in the absence of morphological studies (my own opinion). On the other hand, the authors are largely known molecular biologists, (please see References), and their experimental results, hypotheses and opinions are undoubtedly accepted by me and by the international scientific society.

Thank you for your kind review and advice. The laser scanning electron microscope technique is available at the University of Pécs, and it would be very interesting to analyze the cells at this level, but, of course, we have to discuss these experiments in detail with the experts in this field. These additional morphological experiments would add helpful information to our results, but we feel, that we could answer the basic questions of our hypothesis in this paper, and the cytological results would need another article to prepare.

All changes can be followed by the track change mode.
